# Growth Performance of Buffalo Calves in Response to Different Diets with and without *Saccharomyces cerevisiae* Supplementation

**DOI:** 10.3390/ani14081245

**Published:** 2024-04-22

**Authors:** Fabio Zicarelli, Piera Iommelli, Nadia Musco, Metha Wanapat, Daria Lotito, Pietro Lombardi, Federico Infascelli, Raffaella Tudisco

**Affiliations:** 1Department of Veterinary Medicine and Animal Production, University of Naples Federico II, 80137 Naples, Italy; fabiozicarelli@gmail.com (F.Z.); piera.iommelli@unina.it (P.I.); daria.lotito@unina.it (D.L.); pietro.lombardi@unina.it (P.L.); federico.infascelli@unina.it (F.I.); tudisco@unina.it (R.T.); 2Tropical Feed Resources Research and Development Center (TROFREC), Department of Animal Science, Faculty of Agriculture, Khon Kaen University, Khon Kaen 40002, Thailand; metha@kku.ac.th

**Keywords:** buffalo calves, *Saccharomyces cerevisiae*, growth performance

## Abstract

**Simple Summary:**

Buffalo breeding in Italy has significantly increased in the new century because of mozzarella cheese, but also because of a growing interest in meat production. The nutritional factors that significantly affect animal performance are the forage/concentrate ratio, diet energy and protein content, supplementations (i.e., vitamins, minerals, additives), and the feeding system used. Since antibiotics were banned for auxinic purposes (Reg. 1831/2003/EC), the interest in the potential ability of feed additives to enhance nutrient utilization and animal performance by acting on rumen metabolism has increased. Among such additives, yeast cultures of *Saccharomyces cerevisiae* (SC) have found great interest for application in ruminant nutrition. Therefore, the growth performances of buffalo calves that were fed diets characterized by different forage/concentrate ratio, with or without *Saccharomyces cerevisiae* supplementation, were evaluated in this study.

**Abstract:**

The aim of the present trial was to evaluate the growth performance of buffalo calves fed on diets characterized by different forage/concentrate ratios, with or without *Saccharomyces cerevisiae* supplementation (CBS 493.94, Yea-Sacc^®^). Twenty-four male buffalo calves (mean age of 145.1 ± 16.1 days; mean weight of 108.0 ± 18.7 kg) were assigned randomly to 4 groups, homogeneous in age, that were fed four different diets: diet 1, F:C ratio 50:50; diet 2, F:C ratio 30:70; diet 3, F:C ratio 50:50 + Yea-Sacc^®^; and diet 4, F:C ratio 30:70 + Yea-Sacc^®^. Buffalo calves were individually weighted before the start of the experiment and the data were used as a covariate, being taken monthly until the end of the trial. Dry matter intake (DMI), daily weight gain (DWG) and feed conversion ratio (FCR) were calculated. The differences in diets composition significantly (*p* < 0.01) affected all these parameters. In particular, the animals fed diet 1 and diet 3 showed higher values of DWG (0.91 and 0.88 g/d vs. 0.68 and 0.66 for group 2 and 4) and DMI (5.8 and 5.3 kg/d, respectively) compared to the other groups (4.3 and 4.4 kg/d for group 2 and 4), as well as a higher final body weight (370.5 and 334.1 kg for group 1 and 3 vs. 272.8 and 273.1 kg of group 2 and 4, respectively). Indeed, the supplementation with Yea-Sacc^®^ at the dosage of 1 × 10^E8^ did not affect buffaloes’ growth performance.

## 1. Introduction

In recent years, the interest in buffalo breeding in Italy has increased significantly thanks to the main product derived from buffalo milk, Mozzarella di Bufala Campana, which received the status of Protected Designation of Origin (DOP) in 1996. As a consequence, an increase in the buffalo population from around 200,000 heads in 2000 to over 400,000 in 2020 was reported by ANASB [1,2]. Furthermore, buffalo meat has gained more and more popularity in recent years because of the nutraceutical properties highlighted by some studies [3,4,5]. FAOStat data from 2019 showed an increase of 16% in consumption over the last decade, mainly due to its low fat and cholesterol content [6], for which it has been defined “the healthiest meat among red meats intended for human consumption” [7]. Meat quality, as well as animal performance, are strongly influenced by dietary factors (energy, protein quality, forage/concentrate ratio, supplementations, and feeding systems). Concerning protein quality, Terramoccia et al. [8] and Iommelli et al. [9] reported a better degradation of crude protein (CP) in buffaloes compared to bovines. The higher digestibility of roughage by buffaloes compared to cattle has been reported in the literature [10,11].

The ban of the use of antibiotics for auxinic purposes [12] has led to an increasing interest in those feed additives that may modulate rumen metabolism, enhancing nutrient utilization and animal performance [13]. Starting in 1950, antibiotics have been routinely used in intensive farming to increase farm productivity by improving animals’ health [14]. Unfortunately, the use of antibiotics exerts selective pressure on bacterial populations, leading to antibiotic resistance [14,15], which is now considered a major threat to human and animal health. Antimicrobial resistance has emerged globally, with consequent concerns for both veterinary and human medicine [16,17]. In fact, the intensive use of antibiotics in food-producing animals leads to the diffusion of antibiotic resistant bacteria to humans through food products, animals, or the environment.

Yeast cultures of *Saccharomyces cerevisiae* (SC) have found great interest in ruminant nutrition [18]. SC is able to grow rapidly in the rumen and to facilitate fiber digestion. The micro-nutrients present in SC also stimulate cellulolytic bacteria growth. In addition, SC protects ruminal fermentation from lactic acid accumulation [19]. According to the theory proposed by Newbold et al. [13], in the rumen environment, SC can utilize the remaining dissolved oxygen, saving anaerobic microorganisms from the toxic effects of oxygen, finally resulting in a higher digestion rate and a better growth performance [20]. Indeed, reports on the performance responses of ruminants fed on yeast cultures are controversial. Growth performances were similar or reduced according to Mutsvangwa et al. [21] and Kamra et al. [22], whereas other authors reported an increase in weight gain, feed intake, and feed conversion rate after yeast supplementation [22,23]. To the best of our knowledge, no adverse effects have been reported in the literature for SC use in buffalo.

Weaning represents a critical period for calves due to multifactorial stress including nutritional, physical, and psychological factors, which exert several negative effects on performance, including an increase in the mortality rate of calves [24]. Scientific data regarding the growth and physiological response of buffalo calves fed different diets during and after weaning are scarce. It is difficult to compare the studies in which the growth and physiological responses of buffalo calves to dietary treatments are evaluated. This is mainly because the term “buffalo calf” is used for animals with a body weight ranging between 40 and 220 kg [25].

In such contest, the aim of the present trial was to evaluate the growth performance of buffalo calves fed diets characterized by different forage/concentrate ratios, with or without *Saccharomyces cerevisiae* supplementation. In particular, we tested different F:C ratios because this parameter greatly affects growth, while SC was chosen in view of its low negative effects.

## 2. Materials and Methods

### 2.1. Study Site

The experiment was conducted at the Regional Experimental buffalo farm “Improsta” located in Eboli, Salerno province (145 m s.l. 40°37′1″ N, 15°3′23″ E). The site is characterized by a Mediterranean sub-continental climate, with an annual mean temperature of 15.2° C and an average annual rainfall of 842 mm.

The trial was performed from March to October 2019, in accordance with the Animal Welfare and Good Clinical Practice (Directive 2010/63/EU), and was approved by the local Bioethics Committee (protocol number: 2019/0013729 of February 2019).

### 2.2. Experimental Diets

Two experimental diets were formulated and administered to the buffalo calves, with or without the supplementation of a commercial product Yea-Sacc^®^ (Alltech Inc., Dunboyne, Co., Meath, Ireland), a yeast culture of *Saccharomyces cerevisiae* CBS 493.94. Where applicable, this was added to the diets in ratio of 1 × 10^E8^, as suggested by the manufacturer. The diets were formulated as follows:(1)F:C ratio 50:50;(2)F:C ratio 30:70;(3)F:C ratio 50:50 + Yea-Sacc^®^;(4)F:C ratio 30:70 + Yea-Sacc^®^.

In Table 1, the diets characteristics are reported. *Saccharomyces cerevisiae* was added daily to 500 g of concentrate.

### 2.3. Chemical Composition

Samples (1 kg) of each diet were collected monthly before feeding and analyzed according to AOAC [26] procedures. In particular, diets were milled to pass through a grid of 1.1 mm and analyzed to assess dry matter (DM), crude protein (CP) and ether extract (EE) contents (ID number: 2001.12, 978.04, 920.39 and 978.10, 930.05, respectively). Moreover, structural carbohydrates fractions, neutral detergent fiber (NDF), acid detergent fiber (ADF), and acid detergent lignin (ADL) were analysed according to the work of Van Soest et al. [27]. The starch content was determined through polarimetric detection (Polax L, Atago, Tokyo, Japan), as suggested by the official procedure [28]. The physically effective NDF (peNDF), useful for guaranteeing adequate ruminal activity and the maximun effectiveness of rumen function, was measured with the support of Penn State Particle Separator (PSPS). This consists of 3 meshes of 19 mm, 8 mm, and 4 mm. The peNDF concentration was evaluated by considering the percentage of particle fraction retained (greater than 4 mm) multiplied by the percentage of NDF in the total-mixed-ratio (TMR) sample [29]. UFL (forage unit for lactation) was calculated according to the INRA equation [30].

### 2.4. Animals

Twenty-four male buffalo calves (mean age 145.1 ± 16.1 days; mean weight 108.0 ± 18.7 kg) were recruited. The animals were divided into 4 groups, homogeneous for age, and randomly assigned to different dietary treatments. The feeding of the animals was carried out via TMR (in ratio of 2% of body weight) once a day at 9:00 AM. All animals were housed in well-ventilated sheds, provided with individual feeding and watering arrangements, and dewormed and vaccinated according to the farm’s protocol before the start of the experiment. The trial lasted for 240 days, and the dry matter intake (DMI) was registered daily based on the difference between the feed offered and refusals. Buffalo calves were individually weighted before the start of the experiment and the data were used as a covariate and taken successively each month until the end of the trial. In addition, daily weight gain (DWG) and feed conversion ratio (FCR) were calculated.

### 2.5. Statistical Analysis

Data were analyzed using a one-way ANOVA with the groups (1, 2, 3 and 4) as the factors. The initial body weight of a buffalo was used as the covariate factor. The comparison between the mean values was performed by using Tukey’s test.
yij = μ + Di + ijε
where y represents the experimental data, μ represents the general mean, D is the diet (i = 1, 2, 4), and ε is the error term.

The differences were considered significant at *p* < 0.05. All the statistical procedures were performed using JMP software (version 14; SAS Institute, Cary, NC, USA).

## 3. Results

In Table 2, the chemical composition of the diets is reported. NDF, peNDF, and ADL were significantly higher (*p* < 0.05) in the diets characterized by low F:C, whereas starch showed an opposite trend. The differences in diet composition affected their energy content (UFL 0.84 for diet 1 and 3 vs. 0.86 for diet 2 and 4, respectively). The peNDF content in all the tested diets was appropriate for growing buffaloes [31].

The dietary treatment significantly affected the DMI, final body weight, and DWG of buffalo calves (Table 3). In particular, animals fed on diets 1 and 3 showed the highest values compared to the other groups (*p* < 0.01). Differences were not detected for FCR. Anyway, the supplementation with Yea-Sacc^®^ at the dosage of 1 × 10^E8^ did not affect the calves’ growth performance.

## 4. Discussion

In this trial, the effects of two different diets, characterized by different forage/concentrate ratios were investigated with or without the supplementation of *Saccharomyces cerevisiae* strain CBS 493.94. Results showed that the diets with the higher energy levels (diets 2 and 4) were able to increase dry matter intake, final body weight, and daily weight gain. Comparative studies performed on the digestive physiology and nutritional needs of buffalo have highlighted a greater capacity for fiber utilization compared with cattle and sheep, thus resulting in a better utilization of diets characterized by high-complexity structural carbohydrates [31]. Moreover, in vitro studies demonstrated a higher level of organic matter utilization by the rumen microorganism in buffalo than in bovine [32,33,34,35,36]. Despite that, in this trial, an improvement of buffaloes’ growth performance was observed in the groups fed on the diets characterized by high energy values and a lower forage/concentrate ratio. This result is in agreement with those of Abdel Raheem et al. [37], who compared four different diets characterized by F:C ratios of 80:20, 75:25, 60:40, and 55:45. These authors found an increase in dry matter intake, daily weight gain, and final body weight by increasing the concentrate percentage in the diet. They hypothesized that the increase in DMI could be ascribed to the higher palatability of the concentrate compared to the roughage. Also, DMI was strongly influenced by dietary NDF. It is known that forages must constitute at least 40% of the ruminant diet if the rumen is to maintain adequate functionality and physiology [38]. The high fiber content is the main nutritional difference between forages and concentrates, resulting in a lower energy value of forages. Due to the high forage content in ruminant feed, optimizing forage particle size is a significant feeding strategy for improving forage utilization in ruminants [39]. Indeed, it has been well documented that increasing peNDF content in the diet increases the time spent ruminating and chewing [40,41], with a positive effect on rumen pH and a reduction in the risk of sub-acute and acute ruminal acidosis [39]. Llonch et al. [42] reported that a percentage of peNDF between 6.4% and 15.4% in the diet of beef calves led to a linear increase in daily rumination time. In our trial, the peNDF content of all the diets was appropriate for growing buffaloes [31].

In Figure 1 (panel A, B and C), the DWGs along the experimental period are reported. Panel A compares diets which ar characterized by different F:C ratios. High differences were observed in the first growing period: the higher forage-to-concentrate ratio resulted in a very low DWG (0.230 g) compared to the other diet. Panel B and C show the influence of the addition of SC on both high- and low-F:C diets. SC improved the growth performance of buffalo calves fed on diet 3, but not of those fed on diet 4. This result confirms the possible effect of SC in terms of stimulating the activity of ruminal cellulolytic bacteria. A diet with both SC supplementation and a 30:70 F:C ratio produced a high DWG (0.90 kg) according to Mutsvangwa et al. [21]. However, a mean DWG of 0.68 kg was reported by Infascelli et al. [3].

The inclusion of *Saccharomyces cerevisiae* in the diets has been reported to improve feed intake starting from weaning, which it does by stabilizing ruminal pH and improving fiber digestion, and to stimulate the growth (directly or indirectly) of ruminal cellulolytic bacteria [43]. In our trial, SC did not significantly affect growth performance in buffaloes. Contrasting results are reported in the literature. In studies carried out on lactating buffaloes, some authors highlighted differences in the production and composition of milk [21,44,45], while others did not find any difference due to the inclusion of SC in the diet [41].

Gamal et al. [46] found an increase in final body weight, daily weight gain, and feed conversion ratio (FCR), but differences were not found in DMI in buffalo calves fed on diets supplemented with SC (in ratio of 1%/kg) compared to the control. The higher growth rate found in animals fed on the yeast-supplemented diets may be ascribed to an increased flow of microbial protein leaving the rumen and to a higher supply of amino acids in the small intestine, as suggested by NagamalleswaraRao et al. [47]. These results are in agreement with those of Saha et al. [48] and Kumar and Ramana [24], who showed significant improvements in groups fed on yeast-culture-added diets. Moreover, Kumar and Ramana [24] reported a higher DMI (*p* > 0.05) in calves fed on SC culture (CNCM I-1077 strain in ratio of 0.25 g/head/day)-supplemented diets compared with the control group. Mutsvangwa et al. [21] found significantly greater dry matter intake in bulls after the supplementation of Yea-Sacc^®^ compared to control. Despite a similar average daily gain between groups, those authors reported that the FCR efficiency was not significantly improved by the supplementation (*p* > 0.05), in accordance with our results. On the contrary, Kamra et al. [22] found no difference in the body weight gain, feed intake, feed conversion efficiency in calves fed diets supplemented with yeast cell suspensions (10 mL containing 5 × 10^9^ cells/mL) of *Saccharomyces cerevisiae* (strain ITCCF 2094). It is likely that the differences in the kind of yeast, as well as the dosage, the experimental conditions and the physiological periods of the animals were responsible for the contradictory reports found in the literature. In particular, the lack of significant results on growth performance in this trial could be ascribable to the F:C ratio, since even a 50% of concentrate can be too high, leading the SC to influence fiber digestion.

## 5. Conclusions

This trial showed that the dietary forage/concentrate ratio affects buffalo calves’ growth performance in buffalo calves. On the contrary, no differences were found when supplementing the diet with a commercial product based on the *Saccharomyces cerevisiae* strain CBS 493.94 in a ratio of 1 × 10^8^. Further studies are needed to better define the optimal amount and time of supplementation needed to achieve optimal results.

## 6. Limitations

This study had some limitations: (1) the number of groups was limited (only 2 F:C ratio); and (2) only the SC dosage suggested by the manufacturer was tested.

## Figures and Tables

**Figure 1 animals-14-01245-f001:**
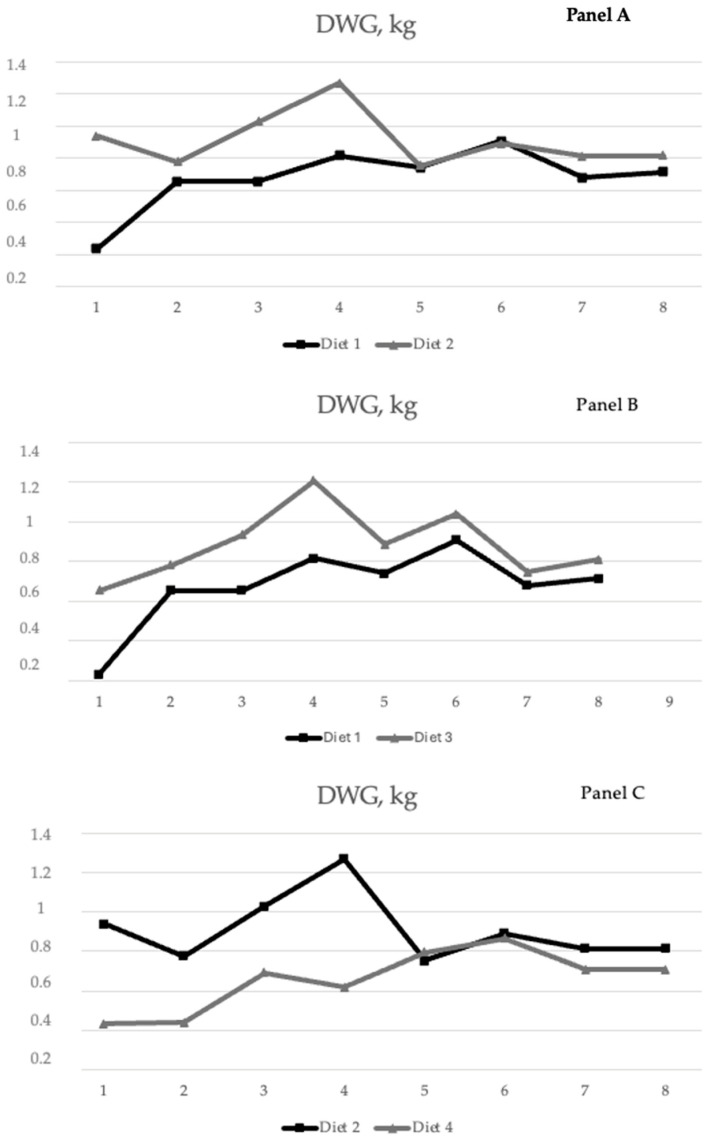
Daily weight gain of buffalo calves fed the four different diets. Panel (**A**): Diet 1 vs. Diet3; Panel (**B**): Diet 1 vs. Diet 3; Panel (**C**): Diet 2 vs. Diet 4. Diet 1: F:C ratio 50:50; Diet 2: F:C ratio 30:70; Diet 3: F:C ratio 50:50 + Yea-Sacc^®^; Diet 4: F:C ratio 30:70 + Yea-Sacc^®^; DWG: daily weight gain.

**Table 1 animals-14-01245-t001:** Ingredients and chemical composition of the experimental diets.

Diet	Unit	1	2	3	4
Supplementation				Yea-Sacc^®^	Yea-Sacc^®^
F:C Ratio		50/50	30/70	50/50	30/70
Ingredients					
Wheat bran	%	30.7	61.2	30.7	61.2
Corn meal	%	18.4	9.2	18.4	9.2
Aalfalfa hay	%	49.2		49.2	
Mixed hay *	%		27.6		27.6
VMS1 **	%	1.7		1.7	
VMS2 ***	%		2.0		2.0
Yea-Sacc^®^	%			0.23	0.23

Diet 1: F:C ratio 50:50; Diet 2: F:C ratio 30:70; Diet 3: F:C ratio 50:50 + Yea-Sacc^®^; Diet 4: F:C ratio 30:70 + Yea-Sacc^®^. * *Phleum Pratense* L., *Lolium italicum* L., *Trifolium pratense* L. ** VMS1: vitamin–mineral supplementation (0.46% vitamin mix ADE: Vit. A: 8.000.000 UI, Vit D: 200.000 UI, Vit E: 3.000 mg; 0.46% buffer mix: calcium: 25%, phosphorus: 1%; 0.77% phosphorus mix: calcium: 5%, phosphorus: 15%). *** VMS2: vitamin–mineral supplementation (0.46% vitamin mix ADE: Vit. A: 8.000.000 UI, Vit D: 200.000 UI, Vit E: 3.000 mg; 1.53% buffer mix: calcium: 25%, phosphorus: 1%).

**Table 2 animals-14-01245-t002:** Chemical composition of the experimental diets (mean ± SD).

Diet	Unit	1	2	3	4
Supplementation				Yea-Sacc^®^	Yea-Sacc^®^
Chemical Composition					
CP	% of DM	16.0 ± 0.6	15.8 ± 0.7	16.0 ± 0.6	15.8 ± 0.7
NDF	% of DM	46.4 ± 6.4 a	42.4 ± 5.0 b	46.4 ± 6.4 a	42.4 ± 5.0 b
ADF	% of DM	33.6 ± 3.6	30.2 ± 3.4	33.6 ± 3.6	30.2 ± 3.4
ADL	% of DM	11.3 ± 0.7 a	8.4 ± 0.5 b	11.3 ± 0.7 a	8.4 ± 0.5 b
EE	% of DM	3.6 ± 0.2	3.2 ± 0.1	3.6 ± 0.2	3.2 ± 0.1
peNDF	% of DM	57.8 ± 3.7 a	50.6 ± 3.5 b	57.8 ± 3.7 a	50.6 ± 3.5 b
Starch	% of DM	20.8 ± 2.5 b	22.4 ± 2.1 a	20.8 ± 2.5 b	22.4 ± 2.1 a
UFL	% of DM	0.84	0.87	0.84	0.87

Diet 1: F:C ratio 50:50; Diet 2: F:C ratio 30:70; Diet 3: F:C ratio 50:50 + Yea-Sacc^®^; Diet 4: F:C ratio 30:70 + Yea-Sacc^®^. CP: crude protein; NDF: neutral detergent fiber; ADF: acid detergent fiber; ADL: lignin detergent fiber; EE: ether extract; peNDF: physically effective NDF; NFE: nitrogen-free extract; UFL: feed units for lactation. a, b: values on the same row with different superscripts differ (*p* < 0.05).

**Table 3 animals-14-01245-t003:** Growth performance of buffalo calves fed on the experimental diets.

	DWG, g/d	Initial, kg	Final Body Weight, kg	DMI, kg/d	FCR
Group					
1	0.66 ± 0.11 B	98.8 ± 5.6	274.1 ± 18.4 B	4.4 ± 0.6 B	6.67 ± 0.04
2	0.88 ± 0.09 A	101.4 ± 8.4	334.1 ± 20.5 AB	5.3 ± 0.5 A	6.02 ± 0.03
3	0.68 ± 0.09 B	94.0 ± 7.6	272.8 ± 16.7 B	4.3 ± 0.6 B	6.32 ± 0.06
4	0.91 ± 0.12 A	137.8 ± 12.4	378.5 ± 24.2 A	5.8 ± 0.3 A	6.37 ± 0.08
RMSE	0.21	6.10	27.79	0.313	0.561

Group 1: F:C ratio 50:50; Group 2: F:C ratio 30:70; Group 3: F:C ratio 50:50 + Yea-Sacc^®^; Group 4: F:C ratio 30:70 + Yea-Sacc^®^. DWG: daily weight gain; DMI: dry matter intake; FCR: feed conversion ratio. A, B: values on the same row with different superscripts differ (*p* < 0.01). RMSE: root-mean-square error.

## Data Availability

Data are contained within the article.

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
