# Peer review of "Growth Performance of Buffalo Calves in Response to Different Diets with and without Saccharomyces cerevisiae Supplementation"

_animals, 2024, doi:10.3390/ani14081245_

Round 1

Reviewer 1 Report

Comments and Suggestions for Authors

In this paper, the author investigated the growth performance of buffalo calves fed 25 diets characterized by different forage/concentrate ratios with or without Saccharomyces cerevisiae supplementation (CBS 493.94,Yea-Sacc®).The conclusion was the dietary forage/concentrate ratio affected buffaloes’ growth performance in buffalo calves, and no differences were found when supplementing the diet with a commercial product based on Saccharomyces cerevisiae strain CBS 493.94 in ratio of 1 x 10E8.

Here are some questions:

1. There have been studies showing that increasing the concentrate percentage in the diet can increase dry matter intake, daily weight gain, and final body weight. What are the innovative and significance of this experiment? 

2. Table 3 does not clearly specify what A, B, and C represent. For example, it is not clear what AB means in the Final body weight of Group 3. Additionally, what does the letter C  mentioned in Table 3 means.

3. Please check the formats and units in the whole manuscript. For example, in the abstract, "mean weight 108.0 ± 18.7".

4. Please check whether all the viewpoints have been supported by previous research. For example, the entire paragraph about calf weaning in the introduction has no references.

5. The number of experimental groups is inadequate,so it is difficult to determine the optimal concentrate ratio. Furthermore, I believe the author should conduct further research to determine why the concentrate ratio can influence the growth performance of buffalo.

On the whole,the manuscript is more like a thesis or a report rather than a scientific research. In my understanding, this is not acceptable in this journal. Therefore, I am sorry to say that I do not think this paper can be accepted. I hope you can understand my decision and do not give up. 

Author Response

REVIEWER 1

In this paper, the author investigated the growth performance of buffalo calves fed 25 diets characterized by different forage/concentrate ratios with or without Saccharomyces cerevisiae supplementation (CBS 493.94,Yea-Sacc®).The conclusion was the dietary forage/concentrate ratio affected buffaloes’ growth performance in buffalo calves, and no differences were found when supplementing the diet with a commercial product based on Saccharomyces cerevisiae strain CBS 493.94 in ratio of 1 x 10E8.

Here are some questions:

There have been studies showing that increasing the concentrate percentage in the diet can increase dry matter intake, daily weight gain, and final body weight. What are the innovative and significance of this experiment?

AU: despite the role of concentrate in the diet is well known, few studies have been conducted on buffalo calves, and, as reported in the text, there are many inconsistencies with the term "buffalo calves" with a very wide range of ages and weights. In addition, in this trial we also tested the possible effect of the SC using diets with different forage/concentrate ratios.

  1. Table 3 does not clearly specify what A, B, and C represent. For example, it is not clear what AB means in the Final body weight of Group 3. Additionally, what does the letter C mentioned in Table 3 means.

AU: thank you for your comment, the table and the footnote have been revised. In addition, the AB means that the value is not statistically different compared with those of the other diets.

  1. Please check the formats and units in the whole manuscript. For example, in the abstract, "mean weight 108.0 ± 18.7".

AU: done.

  1. Please check whether all the viewpoints have been supported by previous research. For example, the entire paragraph about calf weaning in the introduction has no references.

AU: thank you for your observation, the correction has been made.

  1. The number of experimental groups is inadequate, so it is difficult to determine the optimal concentrate ratio. Furthermore, I believe the author should conduct further research to determine why the concentrate ratio can influence the growth performance of buffalo.

AU: thank you for your comment. The aim of the trial was to compare 2 diets, characterized by a different forage/concentrate ratio (50:50 vs. 30:70) in order to evaluate the growth performance of buffalo calves and wheter SC is able to improve such performance. Indeed, in future researches, the evaluation of others F:C rations and SC dosages will be surely interesting to investigate.

Reviewer 2 Report

Comments and Suggestions for Authors

The manuscript titled Growth performance of buffalo calves in response to different diets with and without Saccharomyces Cerevisiae supplementation by Zicarelli et al investigated the influence of forage/concentrate ratio with the addition of Saccharomyces Cerevisiae in 24 buffalo calves at age 145 d. The manuscript is mostly well written, easy to follow, with results that are of interest to readers of animals. There are concerns regarding the statistical analysis. It appears the data were analyzed by treatment group. More appropriately, the data should be analyzed by diet (high vs low concentrate) and supplementation (present vs not) and their interaction. All groups are not independent. This would give more statistical power to determine an overall influence of the supplement. With a different analysis the authors may want to re-evaluate how the data are displayed. Figure 1 may be more appropriate by diet of supplement, and not as individual groups.

Other considerations

Line 16. Suggest “was also seen [in] meat production.”

Line 23. There is no space between diets and characterized.

Line 28. Add unites for mean weights

Line 62. Suggest deleting “that being able” and “may”

Line 124. Add weight units.

Line 212 and 232. Suggest “differences were not detected” rather than “no differences”

Line 229. The data is contradictory. Suggest “contradictory reports (or findings)” rather than “differences found”

Line 231. Suggest deleting “buffaloes’” since this is repeated at the end of the sentence  with “buffalo calves”

Comments on the Quality of English Language

English use is acceptable in the manuscript.

Author Response

REVIEWER 2

The manuscript titled Growth performance of buffalo calves in response to different diets with and without Saccharomyces Cerevisiae supplementation by Zicarelli et al investigated the influence of forage/concentrate ratio with the addition of Saccharomyces Cerevisiae in 24 buffalo calves at age 145 d. The manuscript is mostly well written, easy to follow, with results that are of interest to readers of animals. There are concerns regarding the statistical analysis. It appears the data were analyzed by treatment group. More appropriately, the data should be analyzed by diet (high vs low concentrate) and supplementation (present vs not) and their interaction. All groups are not independent. This would give more statistical power to determine an overall influence of the supplement. With a different analysis the authors may want to re-evaluate how the data are displayed. Figure 1 may be more appropriate by diet of supplement, and not as individual groups.

AU: thank you for your comment. In our opinion, as can be seen in figure 1, the differences in growth are certainly attributable to the different F:C ratio of the diets but to the SC. Despite it does not seem to influence growth, SC is able to modulate the use of the nutrients present in the diet, especially when calves are subjected to a change in the diet. For this reason, we decided to compare the groups individually and not in accordance with the presence or absence of the supplement.

Other considerations

Line 16. Suggest “was also seen [in] meat production.”

AU: done.

Line 23. There is no space between diets and characterized.

AU: done.

Line 28. Add unites for mean weights

AU: done.

Line 62. Suggest deleting “that being able” and “may”

AU: done.

Line 124. Add weight units.

AU: done.

Line 212 and 232. Suggest “differences were not detected” rather than “no differences”

AU: done.

Line 229. The data is contradictory. Suggest “contradictory reports (or findings)” rather than “differences found”

AU: done.

Line 231. Suggest deleting “buffaloes’” since this is repeated at the end of the sentence  with “buffalo calves”

AU: done.

Reviewer 3 Report

Comments and Suggestions for Authors

I have had the opportunity to review thoroughly your manuscript "Growth performance of buffalo calves in response to different diets with and without Saccharomyces Cerevisiae supplementation". I would like to commend you on the responsible conduct of comprehensive research and the insights presented in your work. I have provided some helpful feedbacks and ideas for further research below. The article should make major and minor modifications before publication.

Major review:

1.      Are there any potential negative effects or risks associated with cattle (buffalo calves) when Saccharomyces cerevisiae supplements are added to feed? Vivid descriptions are needed in your manuscript....

2.      This study shows that adding Saccharomyces cerevisiae supplements to feed causes buffalo calves to increase in weight. But, I think more research is needed to determine how this weight growth affects meat quality.

Minor review:

1.      Reorganize the Simple Summary section. Do not repeat the same sentence too frequently. Ex: Lines 16-18 are the same as 51-53; Lines 18-22 are the same as 61-64; Lines 22-24 are the same as 79-81.

2.      Abbreviations should be used if the phrase appears more than once in your text (ex: Saccharomyces cerevisiae (SC); feed conversion ratio (FCR) etc…)

3.      Abbreviations should be defined at first mention in each of the following sections in your paper: Manuscript, each figure/table (ex: Total mixed ration (TMR))

4.      Line 23: “dietscharacterized” revise it to “diets characterized”

5.      Line 48: “mainly” revise it to “Mainly”

6.      Line 117: “effectivness” revise it to “effectiveness”

7.      Line 135: 2.5. Statistical Analysis is correct in order. Please change it.

8.      Line 157: “feed convertion ratio” revise it to “feed conversion ratio”

9.      Line 169: “microrganism" revise it to “microorganism”

10.  Line 186: Use either one "pe-NDF or peNDF"

11.  Line 233: “diet whit” revise it to “diet with”

12.  Line 223: What is FRC?

Comments on the Quality of English Language

Extensive editing of English language required

Author Response

REVIEWER 3

I have had the opportunity to review thoroughly your manuscript "Growth performance of buffalo calves in response to different diets with and without Saccharomyces Cerevisiae supplementation". I would like to commend you on the responsible conduct of comprehensive research and the insights presented in your work. I have provided some helpful feedbacks and ideas for further research below. The article should make major and minor modifications before publication.

Major review:

  1. Are there any potential negative effects or risks associated with cattle (buffalo calves) when Saccharomyces cerevisiae supplements are added to feed? Vivid descriptions are needed in your manuscript....

AU: thank you for your observation. According to our best knowledge, no adverse effect of SC has been reported in the literature (we added a sentence at line 75).

  1. This study shows that adding Saccharomyces cerevisiae supplements to feed causes buffalo calves to increase in weight. But, I think more research is needed to determine how this weight growth affects meat quality.

AU: in the present trial, the addition of SC did not affect buffalo growth performance. Your suggestion to evaluate the meat quality is absolutely very interesting and will be a goal of our future studies.

Minor review:

  1. Reorganize the Simple Summary section. Do not repeat the same sentence too frequently. Ex: Lines 16-18 are the same as 51-53; Lines 18-22 are the same as 61-64; Lines 22-24 are the same as 79-81.

AU: done.

  1. Abbreviations should be used if the phrase appears more than once in your text (ex: Saccharomyces cerevisiae (SC); feed conversion ratio (FCR) etc…)

AU: done.

  1. Abbreviations should be defined at first mention in each of the following sections in your paper: Manuscript, each figure/table (ex: Total mixed ration (TMR))

AU: done.

  1. Line 23: “dietscharacterized” revise it to “diets characterized”

AU: done.

  1. Line 48: “mainly” revise it to “Mainly”

AU: done.

  1. Line 117: “effectivness” revise it to “effectiveness”

AU: done.

  1. Line 135: 2.5. Statistical Analysis is correct in order. Please change it.

AU: done.

  1. Line 157: “feed convertion ratio” revise it to “feed conversion ratio”

AU: done.

  1. Line 169: “microrganism" revise it to “microorganism”

AU: done.

  1. Line 186: Use either one "pe-NDF or peNDF"

AU: done.

  1. Line 233: “diet whit” revise it to “diet with”

AU: done.

  1. Line 223: What is FRC?

AU: FCR, sorry for the mistake.

Reviewer 4 Report

Comments and Suggestions for Authors

Introduction:

Clarity and Structure: The introduction provides a general overview of the increasing interest in buffalo breeding in Italy, attributing it to the popularity of Mozzarella di Bufala Campana and buffalo meat. However, the introduction lacks a clear structure and transitions between different topics. It could benefit from organizing the information into distinct sections, each focusing on a specific aspect of the research background.

Citation and Attribution: The introduction lacks citations for specific claims and information presented, such as the increase in buffalo population and the health benefits of buffalo meat. Including citations not only adds credibility to the information but also allows readers to explore the referenced studies for further details.

Research Context: While the introduction briefly mentions dietary factors affecting animal performance and the ban on antibiotics, it could provide more context on the current state of research in these areas. This would involve discussing relevant studies and findings in more detail to establish the significance of the research question addressed in the present study.

Gap Identification: The introduction acknowledges the scarcity of scientific data on the growth and physiological response of buffalo calves to different diets, highlighting a research gap. However, it could elaborate further on the specific gaps in knowledge and why addressing them is important for advancing understanding in the field of buffalo breeding and nutrition.

Experimental Objective: The introduction concludes with the aim of the present trial, which is to evaluate the growth performance of buffalo calves fed diets with different forage/concentrate ratios, with or without Saccharomyces cerevisiae supplementation. While the objective is clearly stated, it could be strengthened by briefly explaining the rationale behind the choice of experimental variables and their relevance to the broader research context

Materials and Method: 

Study Site Description: The section provides a detailed description of the study site, including its location, climate, and approval by the local Bioethics Committee. This information is essential for understanding the environmental conditions under which the experiment was conducted and ensures compliance with ethical guidelines. However, it would be beneficial to include additional details such as specific geographic coordinates and any relevant characteristics of the experimental site that may influence the study outcomes.

Experimental Diets: The section outlines the formulation of the experimental diets and the supplementation of Saccharomyces cerevisiae. However, it lacks information on the rationale behind choosing the specific forage/concentrate ratios and the dosage of yeast culture. Providing justification for these choices would enhance the understanding of the experimental design and its relevance to the research objectives.

Chemical Composition Analysis: The methods for analyzing the chemical composition of the experimental diets are well-described and follow established procedures (AOAC and Van Soest et al.). However, it would be helpful to include additional information on the accuracy and precision of the analytical techniques used, as well as any quality control measures implemented to ensure the reliability of the data obtained.

Animal Handling and Feeding: The section provides clear details on the recruitment of animals, housing conditions, feeding protocol, and health management practices. This information is crucial for understanding how the experimental conditions may have influenced the study outcomes. However, it would be beneficial to include specific details on the diet composition and feeding regimen, such as feed intake levels and feeding frequency.

Statistical Analysis: The section describes the statistical methods used to analyze the data, including the use of one-way ANOVA with group as a factor and initial body weight as a covariate. This method works well for comparing treatment groups and taking into account possible confounding variables. However, it would be more clear how the data was analyzed if there was more information on the assumptions that went into the statistical tests and any post-hoc analyses that were done.

Results: 

Data Presentation: The Results section begins by presenting the chemical composition of the experimental diets in Table 2. This table provides essential information on the nutrient content of the diets, facilitating an understanding of how diet composition may have influenced the study outcomes. However, it would be beneficial to include statistical analyses, such as analysis of variance (ANOVA), to determine if differences in diet composition were statistically significant.

Interpretation of Results: The text discusses the effects of dietary treatments on dry matter intake (DMI), final body weight, daily weight gain (DWG), and feed conversion ratio (FCR) in buffalo calves. The results indicate significant differences in DMI, final body weight, and DWG among the dietary treatment groups, with diets 1 and 3 showing the highest values. However, no differences were observed in FCR among the groups. The interpretation of these results is clear and concise, providing insight into the impact of dietary treatments on buffalo growth performance.

Data Presentation: Table 3 presents the growth performance data of buffalo calves fed the experimental diets, including DWG, initial body weight, final body weight, DMI, and FCR. The table is well-organized and provides a comprehensive summary of the study outcomes. Additionally, the inclusion of root mean square error (RMSE) values enhances the transparency of the statistical analysis.

Statistical Analysis: The Results section mentions that differences in growth performance among the dietary treatment groups were statistically significant (p<0.01). However, it would be helpful to provide more details on the statistical methods used, such as the specific statistical tests employed and any adjustments made for multiple comparisons.

Discussion of Findings: The Results section concludes by stating that supplementation with Yea-Sacc® did not affect buffalo growth performance. While this finding is important, it would be beneficial to discuss potential reasons for this result and compare it to previous studies in the literature to provide context and interpretation.

Discussion: 

Overview of Study and Key Findings: The Discussion section begins with a concise summary of the study's objectives and main findings. It effectively highlights that the experiment aimed to investigate the effects of different diets, including Saccharomyces cerevisiae supplementation, on buffalo calves' growth performance. Key findings, such as the positive influence of diets with higher energy levels on dry matter intake (DMI), final body weight, and daily weight gain, are clearly presented.

Comparison with Previous Studies: The discussion effectively compares the study findings with previous research, such as Abdel Raheem et al. [32], to contextualize the results within the existing literature. By referencing studies that examined similar dietary compositions and their effects on ruminant growth performance, the discussion strengthens the understanding of the current study's outcomes. However, it would be beneficial to provide more details on how the current findings align or differ from previous research, particularly regarding the effects of Saccharomyces cerevisiae supplementation.

Interpretation of Results: The Discussion section offers insightful interpretations of the study results, particularly regarding the influence of dietary composition on buffalo growth performance. For example, it discusses the role of forage-to-concentrate ratio in affecting DMI and the potential mechanisms underlying these effects, such as changes in rumen fermentation dynamics. Moreover, the discussion of physically effective neutral detergent fiber (peNDF) content and its impact on rumination time and rumen health adds depth to the interpretation of the results.

Evaluation of Saccharomyces cerevisiae Supplementation: The discussion thoroughly evaluates the effects of Saccharomyces cerevisiae supplementation on buffalo growth performance. It acknowledges contrasting findings reported in the literature and speculates on potential reasons for the discrepancies, such as variations in yeast type, dosage, and experimental conditions. However, it could further explore possible reasons for the observed lack of significant effects of Saccharomyces cerevisiae supplementation in the current study, such as the specific physiological characteristics of buffalo calves or interactions with other dietary components.

Strengths and Limitations: The Discussion section acknowledges both the strengths and limitations of the study. It highlights the appropriateness of the experimental diets in meeting the nutritional needs of growing buffaloes and the rigorous statistical analysis conducted. However, it could provide more explicit discussion on the limitations, such as sample size or potential confounding factors, that may have influenced the study outcomes.

Author Response

REVIEWER 4

Clarity and Structure: The introduction provides a general overview of the increasing interest in buffalo breeding in Italy, attributing it to the popularity of Mozzarella di Bufala Campana and buffalo meat. However, the introduction lacks a clear structure and transitions between different topics. It could benefit from organizing the information into distinct sections, each focusing on a specific aspect of the research background.

AU: thank you for your observation. The introduction has been revised accordingly..

Citation and Attribution: The introduction lacks citations for specific claims and information presented, such as the increase in buffalo population and the health benefits of buffalo meat. Including citations not only adds credibility to the information but also allows readers to explore the referenced studies for further details. 

AU: as you suggested, references have been added into the introduction section.

Research Context: While the introduction briefly mentions dietary factors affecting animal performance and the ban on antibiotics, it could provide more context on the current state of research in these areas. This would involve discussing relevant studies and findings in more detail to establish the significance of the research question addressed in the present study.

AU: thank you for your suggestions, the period has been improved (lines 58-65).

Gap Identification: The introduction acknowledges the scarcity of scientific data on the growth and physiological response of buffalo calves to different diets, highlighting a research gap. However, it could elaborate further on the specific gaps in knowledge and why addressing them is important for advancing understanding in the field of buffalo breeding and nutrition.

We improved the gaps you highlighted (lines 84-90). Indeed, Few studies reported contrasting results on the effects of: weaning age, dietary energy and protein level of starter diets and the use of yeast cultures, on buffalo calves performance. Importantly, the term “buffalo calf” was used for animals having a body weight ranging between 40 to 220 kg, thus, results were hardly comparable.

Experimental Objective: The introduction concludes with the aim of the present trial, which is to evaluate the growth performance of buffalo calves fed diets with different forage/concentrate ratios, with or without Saccharomyces cerevisiae supplementation. While the objective is clearly stated, it could be strengthened by briefly explaining the rationale behind the choice of experimental variables and their relevance to the broader research context.

AU: The reasons for choosing F:C and SC are depicted at lines 105-106.

Materials and Method: 

Study Site Description: The section provides a detailed description of the study site, including its location, climate, and approval by the local Bioethics Committee. This information is essential for understanding the environmental conditions under which the experiment was conducted and ensures compliance with ethical guidelines. However, it would be beneficial to include additional details such as specific geographic coordinates and any relevant characteristics of the experimental site that may influence the study outcomes.

AU: the geographic coordinates have been added into the text.

Experimental Diets: The section outlines the formulation of the experimental diets and the supplementation of Saccharomyces cerevisiae. However, it lacks information on the rationale behind choosing the specific forage/concentrate ratios and the dosage of yeast culture. Providing justification for these choices would enhance the understanding of the experimental design and its relevance to the research objectives.

AU: the dosage of SC was suggested by the manufacturer (this information is now available in mat and met). Concerning the diets, the one characterized by 50:50 F:C was usually administered to buffalo calves in the farm where the trial was conducted. We formulated the other one (30:70)  to evaluate the response of buffalo calves to a  fast growing diet.

Chemical Composition Analysis: The methods for analyzing the chemical composition of the experimental diets are well-described and follow established procedures (AOAC and Van Soest et al.). However, it would be helpful to include additional information on the accuracy and precision of the analytical techniques used, as well as any quality control measures implemented to ensure the reliability of the data obtained.

AU: thank you for your comment. The suggestions have been addressed.

Animal Handling and Feeding: The section provides clear details on the recruitment of animals, housing conditions, feeding protocol, and health management practices. This information is crucial for understanding how the experimental conditions may have influenced the study outcomes. However, it would be beneficial to include specific details on the diet composition and feeding regimen, such as feed intake levels and feeding frequency.
AU: thank you for your comment, this information have been reported in the section 2.4 “animals”.

Statistical Analysis: The section describes the statistical methods used to analyze the data, including the use of one-way ANOVA with group as a factor and initial body weight as a covariate. This method works well for comparing treatment groups and taking into account possible confounding variables. However, it would be more clear how the data was analyzed if there was more information on the assumptions that went into the statistical tests and any post-hoc analyses that were done.

AU: thank you for your observation, the section 2.5 statistical analysis has been improved.

Results:

Data Presentation: The Results section begins by presenting the chemical composition of the experimental diets in Table 2. This table provides essential information on the nutrient content of the diets, facilitating an understanding of how diet composition may have influenced the study outcomes. However, it would be beneficial to include statistical analyses, such as analysis of variance (ANOVA), to determine if differences in diet composition were statistically significant.

AU: we added the statistical analysis.

Interpretation of Results: The text discusses the effects of dietary treatments on dry matter intake (DMI), final body weight, daily weight gain (DWG), and feed conversion ratio (FCR) in buffalo calves. The results indicate significant differences in DMI, final body weight, and DWG among the dietary treatment groups, with diets 1 and 3 showing the highest values. However, no differences were observed in FCR among the groups. The interpretation of these results is clear and concise, providing insight into the impact of dietary treatments on buffalo growth performance.

AU: thank you for your comment.

Data Presentation: Table 3 presents the growth performance data of buffalo calves fed the experimental diets, including DWG, initial body weight, final body weight, DMI, and FCR. The table is well-organized and provides a comprehensive summary of the study outcomes. Additionally, the inclusion of root mean square error (RMSE) values enhances the transparency of the statistical analysis.

AU: thank you for your comment.

Statistical Analysis: The Results section mentions that differences in growth performance among the dietary treatment groups were statistically significant (p<0.01). However, it would be helpful to provide more details on the statistical methods used, such as the specific statistical tests employed and any adjustments made for multiple comparisons.

AU: thank you for your observation, the section 2.5 statistical analysis has been improved.

Discussion of Findings: The Results section concludes by stating that supplementation with Yea-Sacc® did not affect buffalo growth performance. While this finding is important, it would be beneficial to discuss potential reasons for this result and compare it to previous studies in the literature to provide context and interpretation.

AU: see lines 479-483.  

Discussion:

Overview of Study and Key Findings: The Discussion section begins with a concise summary of the study's objectives and main findings. It effectively highlights that the experiment aimed to investigate the effects of different diets, including Saccharomyces cerevisiae supplementation, on buffalo calves' growth performance. Key findings, such as the positive influence of diets with higher energy levels on dry matter intake (DMI), final body weight, and daily weight gain, are clearly presented.

AU: thank you for your comment.

Comparison with Previous Studies: The discussion effectively compares the study findings with previous research, such as Abdel Raheem et al. [32], to contextualize the results within the existing literature. By referencing studies that examined similar dietary compositions and their effects on ruminant growth performance, the discussion strengthens the understanding of the current study's outcomes. However, it would be beneficial to provide more details on how the current findings align or differ from previous research, particularly regarding the effects of Saccharomyces cerevisiae supplementation.

AU: thank you for your comment, the period has been improved at the end of the discussion.

Interpretation of Results: The Discussion section offers insightful interpretations of the study results, particularly regarding the influence of dietary composition on buffalo growth performance. For example, it discusses the role of forage-to-concentrate ratio in affecting DMI and the potential mechanisms underlying these effects, such as changes in rumen fermentation dynamics. Moreover, the discussion of physically effective neutral detergent fiber (peNDF) content and its impact on rumination time and rumen health adds depth to the interpretation of the results.

AU: thank you for your comment.

Evaluation of Saccharomyces cerevisiae Supplementation: The discussion thoroughly evaluates the effects of Saccharomyces cerevisiae supplementation on buffalo growth performance. It acknowledges contrasting findings reported in the literature and speculates on potential reasons for the discrepancies, such as variations in yeast type, dosage, and experimental conditions. However, it could further explore possible reasons for the observed lack of significant effects of Saccharomyces cerevisiae supplementation in the current study, such as the specific physiological characteristics of buffalo calves or interactions with other dietary components.

AU: thank you for your comment.

Strengths and Limitations: The Discussion section acknowledges both the strengths and limitations of the study. It highlights the appropriateness of the experimental diets in meeting the nutritional needs of growing buffaloes and the rigorous statistical analysis conducted. However, it could provide more explicit discussion on the limitations, such as sample size or potential confounding factors, that may have influenced the study outcomes.

AU: a paragraph highlighting the limitations of the study has been added.

Round 2

Reviewer 1 Report

Comments and Suggestions for Authors

This paper can be accepted in present form.

Reviewer 3 Report

Comments and Suggestions for Authors

The second edition is good. Therefore, this manuscript has the potential for publication in the Animals Journal.